# Model Checking Fuzzy Computation Tree Logic Based on Fuzzy Decision Processes with Cost

**DOI:** 10.3390/e24091183

**Published:** 2022-08-24

**Authors:** Zhanyou Ma, Zhaokai Li, Weijun Li, Yingnan Gao, Xia Li

**Affiliations:** School of Computer Science and Engineering, North Minzu University, Yinchuan 750000, China

**Keywords:** fuzzy model checking, fuzzy decision processes, fuzzy computation tree logic, cost operator

## Abstract

In order to solve the problems in fuzzy computation tree logic model checking with cost operator, we propose a fuzzy decision process computation tree logic model checking method with cost. Firstly, we introduce a fuzzy decision process model with cost, which can not only describe the uncertain choice and transition possibility of systems, but also quantitatively describe the cost of the systems. Secondly, under the model of the fuzzy decision process with cost, we give the syntax and semantics of the fuzzy computation tree logic with cost operators. Thirdly, we study the problem of computation tree logic model checking for fuzzy decision process with cost, and give its matrix calculation method and algorithm. We use the example of medical expert systems to illustrate the method and model checking algorithm.

## 1. Introduction

Model checking is an important formal verification method. Because of its automatic, model checking has been widely used in the analysis and verification of computer hardware and software systems, communication protocols, security protocols and so on. Model checking is mainly composed of three parts: the first is to model the system under consideration, the second is to use formal language to describe the properties, and the third is to use a model checking algorithm to systematically check whether or not the given model satisfies these properties [1].

Classical model checking [2,3] was formulated for verifying the qualitative properties of systems. However, the Boolean result is not enough for the models with quantitative information, such as a 90 percent probability of the system crashing during operation. At present, more and more complex computer systems have the characteristics of randomness, uncertainty and inconsistency. In order to deal with the verification of complex systems, many quantitative model checking methods have been proposed by academia.

Probabilistic model checking [4,5,6,7,8] mainly deals with the problem of model checking for systems with uncertainties generated by stochastic processes. Its goal is to determine the accuracy of probabilistic systems for quantitative probability specifications. Sometimes models may contain inconsistencies as they connect conflict points or contain components designed by different designers independently. In order to verify complex systems with inconsistencies and uncertainties, multi-valued model checking [9,10,11,12] is proposed. Fuzzy model checking [13,14,15,16,17,18,19,20] pays more attention to the true value of the properties, which is another kind of uncertainty, caused by unclear concept extension [21,22,23]. Both possibility model checking [13,14], and generalized possibility model checking [15,16,24] are based on possibility measure, a combination of possibility measure theory in fuzzy set with model checking. Possibilistic Kripke structure is used to model the system, and possibilistic temporal logic is used to describe properties. Li Yongming et al. [14] use the operators in possibilitic computation tree logic to replace the existence and arbitrary quantifiers of classical computation tree logic to calculate the possibility that the model satisfies the properties. In the process of calculating the possibility measure, the possibility of the path cylinder set after reaching the state also participates in the calculation, but these calculations can be ignored in most systems. Pan Haiyu et al. [19] use fuzzy Kripke structure to model the systems, fuzzy computation tree logic(FCTL) to describe properties and study fuzzy model checking.

Fuzzy Kripke structure is characterized by a state-to-state transition without a cost. However, in daily life, the transition may be different and have some cost [25]. For example, consider the disease diagnosis system studied in [14,19,26]. Suppose there are multi-steps treatments *A* and *B* for a disease, and different treatment needs different costs for each step during the process. The models in the literature [14,19,26] can only describe the situation that experts have been using treatment *A* or *B* during the treatment but cannot describe the situation where experts use *A* as the first and third steps and use *B* as the other steps. In addition, the cost of treatments is also unable to represent and verify. For the above reasons, we have done this paper. First, we define a fuzzy decision process model with a cost function, which can not only describe the nondeterministic choices but also describe the quantitative properties. Second, we introduce the definition of scheduler into the uncertain selection of actions so that the fuzzy decision process with cost function can be transformed into a fuzzy Kripke structure with a cost function. Then, we present the syntax and semantics of fuzzy computation tree logic with the cost operator. Finally, we calculate the quantitative possibility and cost of the problem according to the model checking algorithm. The main contributions of this paper are as follows.

A fuzzy decision process model with cost function is defined, which can describe the cost and other quantitative properties of a fuzzy system. The action property in the model is used to describe the uncertain action selection of the model, and the cost property is used to describe the cost of the system.The FCTL is extended to FCTL with cost operator. The fuzzy computation tree logic with cost operator inherits the existence and arbitrary quantifiers of classical temporal logic and adds operators about cost.The fuzzy computation tree logic model checking quantitative calculation formula and algorithm are given. At the same time, the complexity of the algorithm is analyzed.

The paper is organized as follows: in Section 2, the basic theoretical knowledge of fuzzy mathematics and fuzzy Kripke structure are given. In Section 3, we define a fuzzy decision process model with a cost function. In Section 4, we define fuzzy computation tree logic with a cost operator. In Section 5, we give the fuzzy computation tree logic model checking quantitative calculation formula and algorithm. Section 6 is an example. Section 7 summarizes this paper.

## 2. Preliminaries

A fuzzy set is a mathematical concept proposed by Zadeh in 1965. Fuzziness, in general, refers to any indistinct phenomena, where there is no clear boundary between “stability” and “instability”, “healthy” and “unhealthy”. The transition from one state to another is a continuous process when quantitative changes accumulate and eventually result in a qualitative change, which is due to the uncertainty caused by the breaking of the law of excluded middle. To model and verify fuzzy systems, we provide some necessary knowledge, which includes the fuzzy set, fuzzy set operation, fuzzy matrix operation, closure and others.

**Definition** **1**([27])**.**
*Let X be a universal set. A fuzzy set A of X is a function which associates each element in X a value in the interval [0,1], i.e., A:X⟶[0,1]. For x∈X, A(x) is the membership of x in the fuzzy set A.*

We use F(X) to represent all fuzzy sets in *X*, i.e., F(X)={A∣A:X⟶[0,1]}.

**Definition** **2**([27])**.**
*Let A,B∈F(X), we use A∪B, A∩B, $A^{c}$ to represent the union, intersection and complement of A and B. The definition is as follows.*
*(A∪B)(x)=A(x)∨B(x)=max{A(x),B(x)},*

*(A∩B)(x)=A(x)∧B(x)=min{A(x),B(x)},*

*$A^{c}\left(x\right)=1-A\left(x\right)$.*


Furthermore, we have De Morgan’s laws.$${\left(A\cup B\right)}^{c}=A^{c}\cap B^{c},{\left(A\cap B\right)}^{c}=A^{c}\cup B^{c}.$$

Fuzzy matrix is a kind of special matrix, in which the value of each element is in the interval [0,1]. It has some interesting operations and natures as follows.

**Definition** **3**([28])**.**
*Let R and S be two fuzzy matrixes with m rows and n columns, i.e., $R={\left(r_{ij}\right)}_{m\times n}$, $S={\left(s_{ij}\right)}_{m\times n}$.*
*The standard operations on fuzzy matrixes R,S are defined in the following manner:*

*R=S, if and only if $r_{ij}=s_{ij}$ for all i,j.*

*R⊆S, if and only if $r_{ij}\leq s_{ij}$ for all i,j.*

*$R\cup S={\left(r_{ij}\vee s_{ij}\right)}_{m\times n}$.*

*$R\cap S={\left(r_{ij}\wedge s_{ij}\right)}_{m\times n}$.*

*$R^{c}={\left(1-r_{ij}\right)}_{m\times n}$.*


**Definition** **4**([28])**.**
*Let R be a fuzzy matrix with m rows and n columns, S be a fuzzy matrix with n rows and l columns, i.e., $R={\left(r_{ij}\right)}_{m\times n}$, $S={\left(s_{ij}\right)}_{n\times l}$. The composition operation of R and S is $R\circ S={\left(t_{ij}\right)}_{m\times l}$, where $t_{ij}=\overset{n}{\underset{k=1}{\vee }}\left(r_{ik}\wedge s_{kj}\right)$, (i=1,2,…,m,j=1,2,…,l). For fuzzy matrixes R, S, T the composition operation has some laws.*
(R∘S)∘T=R∘(S∘T);
(R∪S)∘T=(R∘T)∪(S∘T).

Let *X* be a universal set. For the fuzzy matrix $R={\left(R\left(s,t\right)\right)}_{s,t\in X}$, we use $R^{+}$ to denote its transitive closure. When *X* is finite, and *X* has ∣X∣ elements, then $R^{+}=R\cup R^{2}\cup \ldots \cup R^{|X|}$ [29], where $R^{k+1}=R^{k}\circ R$ for any positive integer number *k*. The Kleene closure $R^{*}=R^{0}\cup R^{+}$, for each 1≤s,t≤∣S∣, $R^{0}\left(s,t\right)=\left\{\begin{array}{ccc} 1 & & s=t\\ 0 & & s\neq t \end{array}\right.$.

Transition systems or Kripke structures are the key models for model checking. Corresponding to fuzzy model checking, we expend the notion of fuzzy Kripke structures, defined as follows.

**Definition** **5**([19])**.**
*A fuzzy Kripke structure(FKS) is a tuple K=(S,P,I,AP,L), where**S is a countable, non-empty set of states,**P:S×S⟶[0,1] is the fuzzy transition. For each s∈S, there exist t∈S such that P(s,t)>0,**I:S⟶[0,1] is the initial fuzzy distribution function. The initial state is s, and the truth value is I(s),**AP is the set of atomic propositions,**L:S×AP⟶[0,1] is a fuzzy labeling function. L(s,p) is the truth value to the atomic proposition p in state s.*

The transition of FKS is certain for a pair of states, i.e., P(s,t) is unique. However, on many occasions, we can transmit from one state to another by many methods. In other words, P(s,t) is not certain. We carry out the fuzzy decision processes with a cost which are uncertain in the transition and have a function of cost. For the conditions of daily life, we use the natural number set N as the range of the cost function as an example.

## 3. Fuzzy Decision Processes with Cost

Fuzzy systems are often used to describe the medical expert systems. Due to the different judgment standards of each expert on the patient’s condition and treatment effect, it establishes the model better. At the same time, there are many new problems caused by a variety of treatment options for the same disease. For instance, how to choose the best treatment option in a variety of options? How to evaluate the cost of various treatment options? FKS cannot model the interleaving behavior and the cost of concurrent processes in an adequate manner. For this purpose, we extend FKS to an uncertain system model with cost. The specific definition is as follows.

**Definition** **6**.
*A fuzzy decision process with cost (FDPC) is a tuple $M_{f}=\left(S,Act,P,I,AP,L,C\right)$, where*

*S is a countable, non-empty set of states,*

*Act is the set of actions,*

*P:S×Act×S⟶[0,1] is the fuzzy transition. For each s∈S and α∈Act, there exists t∈S which let P(s,α,t)>0,*

*I:S⟶[0,1] is the initial fuzzy distribution function. For s∈S, the truth value is I(s),*

*AP is the set of atomic propositions,*

*L:S×AP⟶[0,1] is a fuzzy labeling function. L(s,p) is the truth value to the atomic proposition p in state s.*

*C:S×Act⟶N is a cost function. For each s∈S and α∈Act, C(s,α) is the cost of that the action α is selected in state s.*



If *S*, Act and AP are finite, we say the $M_{f}$ is finite. We say that action α is enabled in state *s* if there exists a state t∈S such that P(s,α,t)>0. Act(s) denotes the set of actions which can be enabled in state *s*.

$\overset{\wedge }{\pi }=s_{0}\alpha_{0}s_{1}\alpha_{1}s_{2}\ldots s_{n-1}\alpha_{n-1}s_{n}$ denotes a finite path of $M_{f}$, and $\pi =s_{0}\alpha_{0}s_{1}\alpha_{1}s_{2}\ldots \in {\left(S\times Act\right)}^{\omega }$ denotes an infinite path of $M_{f}$. Paths(s) denotes the set of the infinite paths which begin from state *s*. $Path^{fin}\left(M_{f}\right)$ denotes the set of finite paths which begin from all states of $M_{f}$. $Paths(M_{f})$ is the set of infinite paths which begin from all initial states of $M_{f}$.

**Example** **1**.
*Figure 1, Figure 2, Figure 3 and Figure 4 is a simple, in which there are three transitions represented by α, β and γ. The model has three states $s_{0}$, $s_{1}$ and $s_{2}$. The variables in the state indicate the atomic proposition a and b. Different states have different memberships of atomic propositions. Therefore, we use fuzzy values to describe them. When action $\alpha_{i}$ is used in state $s_{i}$, cost function $C(s_{i},\alpha_{i})$ is generated. When using a single transition α or β or γ, the FKSs are shown in Figure 1, Figure 2 and Figure 3. The FDPC produced by them is shown in Figure 4. The transition possibility is given by the number on the connecting line, and the cost is indicated by the underlined number in the figure. For example, $s_{0}\overset{\alpha ,0.8,\underline{160}}{\to }s_{1}$ indicates that in state $s_{0}$, using treatment scheme α, the possibility of transition to state $s_{1}$ is 0.8, and the treatment cost is 160.*


FDPCs are more complex than FKSs because of the interleaving of the transitions. For example, for a states sequence $s_{0}s_{1}s_{2}$, there is only one possibility in an FKS, but it may be multiple possibilities in a FDPC, such as $s_{0}\alpha s_{1}\beta s_{2}$ or $s_{0}\gamma s_{1}\beta s_{2}$ or the others. We introduce a scheduler to convert FDPC into FKS with cost. In this way, the relevant methods in FKS can be used.

**Definition** **7**.
*Let $M_{f}=\left(S,Act,P,I,AP,L,C\right)$ be a finite FDPC. $Adv:S⟶2^{Act}$ is a function of $M_{f}$. For each s∈S, there is Adv(s)⊆Act(s).*


Under the scheduler Adv, $Path_{Adv}\left(s\right)$ denotes the set of infinite paths which start from the state *s*, $Path_{Adv}^{fin}\left(M_{f}\right)$ denotes the set of finite paths which start from all initial states in $M_{f}$. $Path_{Adv}\left(M_{f}\right)$ denotes the set of infinite paths which start from all initial states in $M_{f}$.

We often care about the maximum (or minimum) possibility. We select the maximum (or minimum) possibility of transition from state *s* to *t* by action in Adv(s) as an example to introduce our thought of scheduler. If there are two or more actions in Adv(s) such that $P\left(s,\alpha_{j},t\right)=P\left(s,\alpha_{k},t\right)=\underset{\alpha_{i}\in Adv\left(s\right)}{\vee }P\left(s,\alpha_{i},t\right)$ for $\alpha_{j},\alpha_{k},\alpha_{i}\in Adv\left(s\right)$, we can select the action by the algebraic product $P\left(s,\alpha_{i},t\right)\cdot C\left(s,\alpha_{i}\right)$. Through the operation of a scheduler, an FDPC can be switched to an FKS with cost.

**Remark** **1**.
*It is easy to prove that the select operations do not change the maximum or minimum possibility of the Adv, because we can use the actions which are selected by us to replace the actions in the maximum or minimum possibility path.*


**Example** **2**.
*For the FDPC of Figure 4, suppose the scheduler function is $Adv\left(s_{0}\right)=\left\{\alpha ,\beta \right\}$, $Adv\left(s_{1}\right)=\left\{\beta ,\gamma \right\}$, $Adv\left(s_{2}\right)=\left\{\gamma \right\}$. The FDPC of Adv, the maximum possibility transition FDPC of Adv and the minimum possibility transition FDPC of Adv are shown in Figure 5, Figure 6 and Figure 7. In fact, Figure 6 and Figure 7 are FKSs with cost.*


The actions are eliminated in the conversion period, so we design an action index matrix to store those actions. The transition matrix in the corresponding FKS and the index matrix for recording actions under a specific scheduler are given below.

Let $M_{f}=\left(S,Act,P,I,AP,L,C\right)$ be a finite FDPC and $\pi =s_{0}\alpha_{0}s_{1}\alpha_{1}s_{2}\ldots \in Paths\left(s\right)$ be a path of $M_{f}$.

$P_{\alpha }$ is a ∣S∣×∣S∣ fuzzy matrix of transition possibility under the action α. For each s,t∈S, $$P_{\alpha }\left(s,t\right)\overset{def}{=}P\left(s,\alpha ,t\right).$$

The left of the equation is the direct transition which transmit from *s* to *t* in FKS which is transmitted by α with cost, but the right is the direct transition which transmit from *s* to *t* by act α in FDPC.

$P_{Adv-max}$ is a ∣S∣×∣S∣ fuzzy matrix which denotes the matrix of maximum transition possibility of Adv. For each s,t∈S,
$$P_{Adv-max}\left(s,t\right)\overset{def}{=}\underset{\alpha \in Adv(s)}{\vee }P\left(s,\alpha ,t\right).$$

$T_{Adv-max}$ is a ∣S∣×∣S∣ action index matrix which records the actions creating the maximum transition possibility. For each s,t∈S, $$T_{Adv-max}\left(s,t\right)\overset{def}{=}\underset{\alpha \in Adv(s)}{argmax}P(s,\alpha ,t).$$

$P_{Adv-min}$ is a ∣S∣×∣S∣ fuzzy matrix which denotes the matrix of minimum transition possibility of Adv. For each s,t∈S, $$P_{Adv-min}\left(s,t\right)\overset{def}{=}\underset{\alpha \in Adv(s)}{\wedge }P(s,\alpha ,t).$$

$T_{Adv-min}$ is a ∣S∣×∣S∣ action index matrix which records the actions creating the minimum transition possibility. For each s,t∈S, $$T_{Adv-min}\left(s,t\right)\overset{def}{=}\underset{\alpha \in Adv(s)}{argmin}P(s,\alpha ,t).$$

We often pay attention to the maximum and minimum possibility of FDPC, but they are the special Adv where Adv(s)≡Act(s) for all s∈S. We use the special symbol $\alpha_{max},\alpha_{min}$ to denote the index of this Adv.

$P_{\alpha_{max}}$ is a ∣S∣×∣S∣ fuzzy matrix which denotes the matrix of maximum transition possibility of FDPC. For each s,t∈S, $$P_{\alpha_{max}}\left(s,t\right)\overset{def}{=}\underset{\alpha \in Act(s)}{\vee }P(s,\alpha ,t).$$

$T_{\alpha_{max}}$ is a ∣S∣×∣S∣ action index matrix which records the actions creating the maximum transition possibility. For each s,t∈S, $$T_{\alpha_{max}}\left(s,t\right)\overset{def}{=}\underset{\alpha \in Act(s)}{argmax}P(s,\alpha ,t).$$

$P_{\alpha_{min}}$ is a ∣S∣×∣S∣ fuzzy matrix which denotes the matrix of minimum transition possibility of FDPC. For each s,t∈S, $$P_{Adv-min}\left(s,t\right)\overset{def}{=}\underset{\alpha \in Act(s)}{\wedge }P(s,\alpha ,t).$$

$T_{\alpha_{min}}$ is a ∣S∣×∣S∣ action index matrix which records the actions creating the minimum transition possibility. For each s,t∈S, $$T_{\alpha_{min}}\left(s,t\right)\overset{def}{=}\underset{\alpha \in Act(s)}{argmin}P(s,\alpha ,t).$$

$C_{\alpha }$ is a ∣S∣×1 matrix declaring the cost activating action α from state *s*, for each s∈S, $$C_{\alpha }\overset{def}{=}C(s,\alpha ).$$

**Example** **3**.
*For the FDPC of Figure 4.*

*The action α transition possibility matrix and the cost matrix activating action α is*

$$P_{\alpha }=\left(\begin{array}{ccc} 0.2 & 0.8 & 0.4\\ 0.1 & 0.3 & 0.5\\ 0.2 & 0.3 & 0.8 \end{array}\right),C_{\alpha }=\left(\begin{array}{c} 160\\ 180\\ 100 \end{array}\right).$$


*The action β transition possibility matrix and the cost matrix activating action β is*


$$P_{\beta }=\left(\begin{array}{ccc} 0.3 & 0.7 & 0.2\\ 0.2 & 0.4 & 0.3\\ 0.1 & 0.5 & 0.4 \end{array}\right),C_{\beta }=\left(\begin{array}{c} 100\\ 115\\ 70 \end{array}\right).$$


*The action γ transition possibility matrix and the cost matrix activating action γ is*


$$P_{\gamma }=\left(\begin{array}{ccc} 0.5 & 0.6 & 0.1\\ 0.4 & 0.7 & 0.2\\ 0.5 & 0.6 & 0.3 \end{array}\right),C_{\gamma }=\left(\begin{array}{c} 68\\ 73\\ 40 \end{array}\right).$$



The maximum transition possibility matrix, maximum transition possibility action index matrix, minimum transition possibility matrix and minimum transition possibility action index matrix is$$P_{\alpha_{max}}=\left(\begin{array}{ccc} 0.5 & 0.8 & 0.4\\ 0.4 & 0.7 & 0.5\\ 0.5 & 0.6 & 0.8 \end{array}\right),T_{\alpha_{max}}=\left(\begin{array}{ccc} \gamma & \alpha & \alpha \\ \gamma & \gamma & \alpha \\ \gamma & \gamma & \alpha \end{array}\right),\;P_{\alpha_{min}}=\left(\begin{array}{ccc} 0.2 & 0.6 & 0.1\\ 0.1 & 0.3 & 0.2\\ 0.1 & 0.3 & 0.3 \end{array}\right),T_{\alpha_{min}}=\left(\begin{array}{ccc} \alpha & \gamma & \gamma \\ \alpha & \alpha & \gamma \\ \beta & \alpha & \gamma \end{array}\right).$$

Under the maximum transition possibility matrix and minimum transition possibility matrix, the FDPC of Figure 4 turns into the FKSs with cost in Figure 8 and Figure 9.

Since the cost is generated in the process of each activation action, in order to solve the cost-related problems in model checking, we determine the cost through a deterministic selection strategy for the action, and then calculate the expected cost. Because model checking pays more attention to the possibility of transition, this paper takes the maximum possibility of one-step transition as the selected strategy. First, we select the action by the above matrix. Then, we select the successor state by the maximum possibility.

**Definition** **8**.
*Let $M_{f}=\left(S,Act,P,I,AP,L,C\right)$ be a finite FDPC and $\pi =s_{0}\alpha_{0}s_{1}\alpha_{1}s_{2}\ldots \in Paths\left(s\right)$ be a path of $M_{f}$. $cost\left[=k\right]\left(\pi \right)\overset{def}{=}C\left(s_{k-1},\alpha_{k-1}\right)$ denotes the instantaneous cost of the step k.*


Under the scheduler Adv, we use each step to choose the maximum possibility to transmit from the current state as an example, and the step *k* instantaneous expected cost is defined as



$Ex_{Adv}^{s}\left(cost\left[=k\right]\right)\overset{def}{=}$





$\left(\underset{0<j\leq k}{\wedge }\left(\underset{t_{j}\in S}{\vee }P_{Adv}\left(t_{j-1},t_{j}\right)\right)\right)\cdot C\left(t_{j-1},\alpha_{k-1}\right).$



We use $cost\left[\leq k\right]\left(\pi \right)\overset{def}{=}\sum_{i=1}^{k}C\left(s_{i-1},\alpha_{i-1}\right)$ to denote the cumulative cost of the first *k* steps. The cumulative expected cost of the first *k* steps is defined as
$$Ex_{Adv}^{s}\left(cost\left[\leq k\right]\right)\overset{def}{=}\sum_{i=1}^{k}Ex_{Adv}^{s}\left(cost\left[=i\right]\right).$$

The previous descriptions are all about the expected cost without limiting the states in the path. However, in the actual process, some restrictions may be added to the states in the path.

$cost\left[⋄F\right]\left(\pi \right)=cost\left(\overset{\wedge }{\pi }\right)\overset{def}{=}\sum_{i=1}^{n}C\left(s_{i-1},\alpha_{i-1}\right)$ denotes the cumulative cost that the path would reach the state in *F*, where F⊆S, $\overset{\wedge }{\pi }=s_{0}\alpha_{0}\ldots s_{n}$ is the prefix of π and $s_{n}\in F$ but the other states in $\overset{\wedge }{\pi }$ is not in *F*. $Ex_{Adv}^{s}\left(cost\left[=n\left(⋄F\right)\right]\right)$ denotes the skep *n* instantaneous expected cost under the scheduler Adv which is the deformation of $Ex_{Adv}^{s}\left(cost\left[=k\right]\right)$, defined as below.$$\begin{array}{c} Ex_{Adv}^{s}\left(cost\left[=1\left(⋄F\right)\right]\right)\overset{def}{=}\left(\underset{t\in F}{\vee }P_{Adv}\left(s,t\right)\right)\cdot C(s,\alpha ),\\ Ex_{Adv}^{s}\left(cost\left[=k\left(⋄F\right)\right]\right)\overset{def}{=}\\ \left(\underset{0<j<k}{\wedge }\left(\underset{t\in S/F}{\vee }P_{Adv}\left(t_{j-1},t_{j}\right)\right)\wedge P_{Adv}\left(t_{k}-1,t\right)\right)\cdot C\left(t_{j-1},\alpha_{k-1}\right). \end{array}$$

$Ex_{Adv}^{s}\left(cost\left[\leq k\left(⋄F\right)\right]\right)$ denotes the first steps *k* cumulative expected cost under the scheduler Adv which reaches *F* in step *k*, defined as below,$$\begin{array}{c} Ex_{Adv}^{s}\left(cost\left[\leq 1\left(⋄F\right)\right]\right)\overset{def}{=}Ex_{Adv}^{s}(cost\left[=1\left(⋄F\right)\right]),\\ Ex_{Adv}^{s}\left(cost\left[\leq k\left(⋄F\right)\right]\right)\overset{def}{=}\\ (\sum_{i=1}^{k-1}(\left(\underset{t\in F}{\vee }\left(\underset{0<j<k}{\wedge }\left(\underset{s\in S/F}{\vee }P_{Adv}\left(t_{j-1},t_{j}\right)\right)\right)\right)\cdot \\ C\left(s_{j-1},\alpha_{j-1}\right))+\left(\left(\underset{t\in F}{\vee }\left(\underset{0<j<k}{\wedge }\left(\underset{s\in S/F}{\vee }P_{Adv}\left(t_{j-1},t_{j}\right)\wedge P_{Adv}\left(s_{k-1},t\right)\right)\right)\right)\cdot C\left(s_{k-1},\alpha_{k-1}\right)\right)). \end{array}$$

$Ex_{Adv}^{s}\left(cost\left[⋄F\right]\right)$ is the cumulative expected cost from *s* to the state in *F*, defined as below,
$$Ex_{Adv}^{s}\left(cost\left[⋄F\right]\right)\overset{def}{=}\underset{1\leq k\leq n}{sup}Ex_{Adv}^{s}(cost\left[\leq k\left(⋄F\right)\right]).$$

Where *n* is the max step of all paths that can reach *F* under the restrictive condition ‘No or one ring’. Why do we have the restrictive condition ‘No or one ring’? Because it has no contribution for transition possibility changing. Without the amount of rings, all states of the path which can reach *F* are less than or equal ∣S/F∣+1, thus we can get that the max skep n≤∣S/F∣.

## 4. FCTL with Cost Operator

We present the FCTL with cost operators in this section, i.e., expand FCTL for our FDPC with cost operator. We expand FCTL [19] with the cost operator. The syntax and semantics are as below.

**Definition** **9**(FCTL syntax)**.**
*The FCTL state formula is defined inductively as follows,*
*

$\Phi ::=true|a|\Phi_{1}\wedge \Phi_{2}|\neg \Phi |\exists \varphi |\forall \varphi |E\left(=k\right)|E\left(\leq k\right)|E(\Phi ),$

*

*where φ is a path formula, a∈AP.*

*Furthermore, the FCTL path formula is,*

*$\varphi ::=◯\Phi |\Phi_{1}⊔\Phi_{2}$,*

*where Φ, $\Phi_{1}$ and $\Phi_{2}$ are state formulas.*


**Definition** **10**(FCTL semantics)**.**
*Let $M_{f}=\left(S,Act,P,I,AP,L,C\right)$ be an finite FDPC, ∣∣Φ∣∣:S⟶[0,1] be a function. For FCTL with cost, the semantic of state formula Φ is defined as follows.*$$\begin{array}{cc} ||true||(s) & =1,\\ ||a||(s) & =L(s,a),\\ ||\Phi_{1}\wedge \Phi_{2}||\left(s\right) & =||\Phi_{1}||\left(s\right)\wedge ||\Phi_{2}||\left(s\right),\\ ||\neg \Phi ||(s) & =1-||\Phi ||(s),\\ ||\exists \varphi |{|}_{Adv}\left(s\right) & =\underset{\pi \in Paths_{Adv}\left(s\right)}{\vee }||\varphi ||\left(\pi \right),\\ ||\forall \varphi |{|}_{Adv}\left(s\right) & =\underset{\pi \in Paths_{Adv}\left(s\right)}{\wedge }||\varphi ||\left(\pi \right),\\ ||E\left(=k\right)|{|}_{Adv}\left(s\right) & =Ex_{Adv}^{s}\left(cost\left[=k\right]\right),\\ ||E\left(\leq k\right)|{|}_{Adv}\left(s\right) & =Ex_{Adv}^{s}\left(cost\left[\leq k\right]\right),\\ ||E\left(\Phi \right)|{|}_{Adv}\left(s\right) & =Ex_{Adv}^{s}\left(cost\left[⋄Sat\left(\Phi \right)\right]\right) \end{array}$$*where*Sat(Φ)={s∈S∣∣∣Φ∣∣(s)>0}.
*For the given scheduler Adv and $\pi \in Paths_{Adv}\left(s\right)$, the semantic of path formula φ is defined as below.*

*$||◯\Phi |{|}_{Adv}=P_{Adv}\left(s_{0},s_{1}\right)\wedge ||\Phi ||\left(s_{1}\right)$,*


$||\Phi_{1}⊔\Phi_{2}|{|}_{Adv}\left(\pi \right)$


*

$=||\Phi_{2}||\left(s\right)\vee \underset{j>0}{\vee }\left(||\Phi_{1}||\left(s_{0}\right)\wedge \underset{k<j}{\wedge }\left(P_{Adv}\left(s_{k-1},s_{k}\right)\right)\wedge ||\Phi_{1}||\left(s_{k}\right)\wedge \left(P_{Adv}\left(s_{j-1},s_{j}\right)\right)\wedge ||\Phi_{2}||(s_{j}).\right)$

*


## 5. Model Checking Fuzzy Computation Tree Logic Based on Fuzzy Decision Processes with Cost

The FCTL model checking problem on FDPC is defined as the following. Given a FDPC $M_{f}$, a state *s* of $M_{f}$, a FCTL state formula Φ and a step *k*, then calculate the true value of state *s* satisfying the state formula or the expect cost. We often consider the maximum and minimum possibilitic truth values. We give the computing method of $||\exists ◯\Phi |{|}_{max}\left(s\right)$, $||\exists ◯\Phi |{|}_{min}\left(s\right)$, $||\forall ◯\Phi |{|}_{max}\left(s\right)$, $||\forall ◯\Phi |{|}_{min}\left(s\right)$, $||\exists \Phi_{1}⊔\Phi_{2}|{|}_{max}\left(s\right)$, $||\exists \Phi_{1}⊔\Phi_{2}|{|}_{min}\left(s\right)$, $||\forall \Phi_{1}⊔\Phi_{2}|{|}_{max}\left(s\right)$, $||\forall \Phi_{1}⊔\Phi_{2}|{|}_{min}\left(s\right)$, $||E\left(=k\right)|{|}_{max}\left(s\right)$, $||E\left(=k\right)|{|}_{min}\left(s\right)$, $||E\left(\leq k\right)|{|}_{max}\left(s\right)$, $||E\left(\leq k\right)|{|}_{min}\left(s\right)$, $||E\left(\Phi \right)|{|}_{max}\left(s\right)$ and $||E\left(\Phi \right)|{|}_{min}\left(s\right)$ as below. There are some useful matrixes and operations given.

Let $M_{f}=\left(S,Act,P,I,AP,L,C\right)$ be a finite FDPC, $D_{\Phi }$ be a ∣S∣×∣S∣ fuzzy diagonal matrix for state formula Φ. For each s,t∈S, $$D_{\Phi }\left(s,t\right)\overset{def}{=}\left\{\begin{array}{ccc} & ||\Phi ||(s) & s=t\\ & 0 & Otherwise \end{array}\right.$$

$P_{\Phi }$ is a ∣S∣×1 fuzzy matrix. For each s∈S, $$P_{\Phi }\left(s\right)\overset{def}{=}||\Phi ||(s).$$

We use $P_{\alpha |max|}$ to realize the transition of only choosing the maximum truth value transition using action α. $P_{\alpha |max|}$ is a ∣S∣×∣S∣ fuzzy matrix and defined as that for each s,t∈S, $$P_{\alpha |max|}\left(s,t\right)\overset{def}{=}\left\{\begin{array}{ccc} & P(s,\alpha ,t) & t=\underset{t\in S}{argmax}f\left(P\left(s,\alpha ,t\right),cost\left(s,\alpha \right)\right)\\ & 0 & Otherwise \end{array}\right.$$where *f* is a function mapping P×C to [0,1]. *f* is decided by the need in usual. In this paper, we set f(P(s,α,t),cost(s,α))=P(s,α,t) if there is only one maximum transition from *s* to *t*. When it is multiple, we select the first maximum transition by elements in the matrix.

Through those matrixes, we can reduce the state transition matrix in FKS to a sparse matrix containing only one-step maximum possibility.

*E* is a ∣S∣×1 fuzzy matrix with all elements equal to 1. We use it to turn the matrix into a vector.

We also use an auxiliary matrix identified as $D_{F}$, for the restricted set *F*. $D_{F}$ is defined below. $$D_{F}=\left(\begin{array}{ccc} D[0,0] & \cdots & D[0,n]\\ \cdots & \vdots & \cdots \\ D[n,0] & \cdots & D[n,n] \end{array}\right),$$ where $D\left[i,j\right]=\left\{\begin{array}{ccc} 1 & & s_{j}\in F\\ 0 & & s_{j}\notin F \end{array}\right.$

$P_{\alpha |F}\overset{def}{=}D_{F}\cap P_{\alpha }$ is the restricted transition matrix under the act α which restrict only transition to *F*. $P_{\alpha |F}^{\prime }\overset{def}{=}\left(D_{S}-D_{F}\right)\cap P_{\alpha }$ is the restricted transition matrix under the act α which cannot transmit to *F*. Using matrixes, we can describe the constraints.

Using our matrixes, we can re-represent the cost in Definition 8.

The first *k* steps cumulative expected cost$$\begin{array}{c} Ex_{Adv}^{s}\left(cost\left[\leq k\right]\right)\\ =\sum_{i=1}^{k}Ex_{Adv}^{s}\left(cost\left[=i\right]\right)\\ =\sum_{i=1}^{k}\left(\left(\underset{0<j\leq i}{\wedge }\left(\underset{t_{j}\in S}{\vee }P_{Adv}\left(t_{j-1},t_{j}\right)\right)\right)\cdot C\left(t_{i-1},\alpha_{i-1}\right)\right)\\ =\sum_{i=1}^{k}\left(\left(\underset{0<j\leq i}{\wedge }\left(\underset{t_{j}\in S}{\vee }\underset{\alpha_{j-1}\in Adv\left(t_{j-1}\right)}{\vee }P_{Adv}\left(t_{j-1},\alpha_{j-1},t_{j}\right)\right)\right)\cdot C\left(t_{i-1},\alpha_{i-1}\right)\right)\\ =\sum_{i=1}^{k}\left(\left(\underset{0<j\leq i}{\wedge }\left(\underset{t_{j}\in S}{\vee }P_{Adv-max}\left(t_{j-1},t_{j}\right)\right)\right)\cdot C\left(t_{i-1},\alpha_{i-1}\right)\right)\\ =\sum_{i=1}^{k}\left(\left(P_{Adv-max|max|}\circ P_{Adv-max|max|}\circ \cdots \circ P_{Adv-max|max|}\circ E\right)\left(s\right)\cdot C_{\alpha_{i-1}}\left(t_{i-1}\right)\right)\\ =\sum_{i=1}^{k}(\left(P_{Adv-max|max|}^{i}\circ E\right)\left(s\right)\cdot C_{\alpha_{i-1}}\left(t_{i-1}\right)). \end{array}$$

The restricted set F⊆S step *k* instantaneous expected cost$$\begin{array}{ll} Ex_{Adv}^{s} & \left(cost\left[=1\left(⋄F\right)\right]\right)\\ & =(\underset{t\in F}{\vee }P_{Adv}\left(s,t\right))\cdot C\left(s,\alpha \right)\\ & =(\underset{t\in F}{\vee }\underset{\alpha \in Adv(s)}{\vee }P\left(s,\alpha ,t\right))\cdot C\left(s,\alpha \right)\\ & =(\left(P_{Adv-max|F|max|}\circ E\right)\left(s\right)\cdot C_{\alpha }\left(s\right)\\ & {Ex}_{Adv}^{s}\left(cost\left[=k\left(⋄F\right)\right]\right)\\ & =\left(\underset{t\in F}{\vee }\left(\underset{0<j<k}{\wedge }\left(\underset{s\in S\setminus F}{\vee }P_{Adv}\left(s_{j-1},s_{j}\right)\right)\wedge P_{Adv}\left(s_{k-1},t\right)\right)\right)\cdot C\left(s_{k-1},\alpha_{k-1}\right)\\ & =\left(\underset{t\in F}{\vee }\left(\underset{0<j<k}{\wedge }\left(\underset{s\in S\setminus F}{\vee }\underset{\alpha_{j-1}\in Adv\left(s_{j-1}\right)}{\vee }P\left(s_{j-1},\alpha_{j-1},s_{j}\right)\right)\wedge \underset{\alpha_{k-1}\in Adv\left(s_{k-1}\right)}{\vee }P\left(s_{k-1},\alpha_{k-1},t\right)\right)\right)\cdot C\left(s_{k-1},\alpha_{k-1}\right)\\ & =(\left(P_{Adv-\max|F|\max|}^{\prime }\circ \cdots \circ P_{Adv-\max|F|\max|}^{\prime }\circ P_{Adv-\max|F|\max|}^{}\circ E\right)\left(s\right)\cdot C_{\alpha_{k-1}}\left(s_{k-1}\right))\\ & =(\left(P_{Adv-\max|F|\max|}^{{}^{\prime }k-1}\circ P_{Adv-\max|F|\max|}\circ E\right)\left(s\right)\cdot C_{\alpha_{k-1}}\left(s_{k-1}\right)) \end{array}$$

The restricted set F⊆S first *k* steps cumulative expected cost



$Ex_{Adv}^{s}\left(cost\left[\leq 1\left(⋄F\right)\right]\right)$





$=Ex_{Adv}^{s}\left(cost\left[=1\left(⋄F\right)\right]\right)$





$=\left(P_{Adv-\max|F|\max|}^{}\circ E\right)\left(s\right)\cdot C_{\alpha_{k-1}}\left(s\right)$





$Ex_{Adv}^{s}\left(cost\left[\leq k\left(⋄F\right)\right]\right)$





$=\sum_{i=1}^{k-1}\left(\left(\underset{0<j\leq i}{\wedge }\left(\underset{s\in S\setminus F}{\vee }P_{Adv}\left(s_{j-1},s_{j}\right)\right)\right)\cdot C\left(s_{j-1},\alpha_{j-1}\right)\right)+\left(\underset{t\in F}{\vee }\left(\underset{0<j<k}{\wedge }\left(\underset{s\in S\setminus F}{\vee }P_{Adv}\left(s_{j-1},s_{j}\right)\right)\wedge P_{Adv}\left(s_{k-1},t\right)\right)\right)\cdot C\left(s_{k-1},\alpha_{k-1}\right)$



$=\sum_{i=1}^{k-1}\left(\left(P_{Adv-\max|F|\max|}^{{}^{\prime }i}\circ E\right)\left(s\right)\cdot C_{\alpha_{i-1}}\left(s_{i-1}\right)\right)+\left(\left(P_{Adv-\max|F|\max|}^{{}^{\prime }k-1}\circ P_{Adv-\max|F|\max|}^{}\circ E\right)\left(s\right)\cdot C_{\alpha_{k-1}}\left(s_{k-1}\right)\right)$
where $P_{\alpha |F|max|}$ is the operation that to $P_{\alpha }$ first count $P_{\alpha |F}$ and second count $P_{\alpha |max|}$ for $P_{\alpha |F}$. $P_{\alpha |F|max|}^{\prime }$ is the operation that to $P_{\alpha }$ first count $P_{\alpha |F}^{\prime }$ and second count $P_{\alpha |max|}$ for $P_{\alpha |F}^{\prime }$.

$||\exists ◯\Phi |{|}_{max}\left(s\right)$ is the maximum truth value of that there exists a path that starts from state *s* and satisfies ◯Φ.
(1)$$\begin{array}{c} ||\exists ◯\Phi |{|}_{\max}\left(s\right)=\left(P_{\alpha_{max}}\circ P_{\Phi }\right)\left(s\right). \end{array}$$

The proof is placed in Appendix A.

$||\exists ◯\Phi |{|}_{min}\left(s\right)$ is the minimum truth value of that there exists a path that starts from state *s* and satisfies ◯Φ.
(2)$$\begin{array}{c} ||\exists ◯\Phi |{|}_{\min}\left(s\right)=\left(P_{\alpha_{min}}\circ P_{\Phi }\right)\left(s\right). \end{array}$$

The proof is placed in Appendix A.

$||\forall ◯\Phi |{|}_{max}\left(s\right)$ is the maximum truth value of that all of the paths which start from state *s* satisfy ◯Φ.
(3)$$\begin{array}{c} ||\forall ◯\Phi ||{}_{\max}\left(s\right)={\left({\left(P_{\alpha_{\max}}\circ D_{\Phi }\right)}^{c}\circ E\right)}^{c}\left(s\right) \end{array}$$

The proof is placed in Appendix A.

$||\forall ◯\Phi |{|}_{min}\left(s\right)$ is the minimum truth value of that all of the paths which start from state *s* satisfy ◯Φ.
(4)$$\begin{array}{c} ||\forall ◯\Phi ||{}_{\min}\left(s\right)={\left({\left(P_{\alpha_{\min}}\circ D_{\Phi }\right)}^{c}\circ E\right)}^{c}\left(s\right) \end{array}$$

The proof is placed in Appendix A.

$||\exists \Phi_{1}⊔\Phi_{2}|{|}_{max}\left(s\right)$ is the maximum truth value of that there exists a path that starts from state *s* satisfies $\Phi_{1}⊔\Phi_{2}$.
(5)$$\begin{array}{c} ||\exists \Phi_{1}⊔\Phi_{2}|{|}_{max}\left(s\right)=\left({\left(D_{\Phi_{1}}\circ P_{\alpha_{\max}}\right)}^{*}\circ P_{\Phi_{2}}\right)\left(s\right) \end{array}$$

The proof is placed in Appendix A.

$||\exists \Phi_{1}⊔\Phi_{2}|{|}_{min}\left(s\right)$ is the minimum truth value of that there exists a path that starts from state *s* satisfies $\Phi_{1}⊔\Phi_{2}$.
(6)$$\begin{array}{c} ||\exists \Phi_{1}⊔\Phi_{2}|{|}_{min}\left(s\right)=\left({\left(D_{\Phi_{1}}\circ P_{\alpha_{\min}}\right)}^{*}\circ P_{\Phi_{2}}\right)\left(s\right) \end{array}$$

The proof is placed in Appendix A.

$||\forall \Phi_{1}⊔\Phi_{2}|{|}_{max}\left(s\right)$ is the maximum truth value of that all of the paths which start from state *s* satisfy $\Phi_{1}⊔\Phi_{2}$.
(7)$$\begin{array}{cc} \; & ||\forall \Phi_{1}⊔\Phi_{2}|{|}_{max}\left(s\right)\\ \; & =(P_{\Phi_{2}}\cup ([{\left({\left(D_{\Phi_{1}}\circ P_{\alpha_{\max}}\right)}^{c}\circ D_{S}\right)}^{c}\circ {\left({\left(D_{S}\circ {\left(D_{\Phi_{1}}\circ P_{\alpha_{\max}}\right)}^{c}\right)}^{c}\right)}^{*}\circ D_{\Phi_{2}}{]}^{c}\\ \; & {\circ E)}^{c})\left(s\right) \end{array}$$

The proof is placed in Appendix A.

$||\forall \Phi_{1}⊔\Phi_{2}|{|}_{min}\left(s\right)$ is minimum the truth values of that all of the paths that start from state *s* satisfy $\Phi_{1}⊔\Phi_{2}$.
(8)$$\begin{array}{cc} \; & ||\forall \Phi_{1}⊔\Phi_{2}|{|}_{min}\left(s\right)\\ \; & =(P_{\Phi_{2}}\cup ([{\left({\left(D_{\Phi_{1}}\circ P_{\alpha_{\min}}\right)}^{c}\circ D_{S}\right)}^{c}\circ {\left({\left(D_{S}\circ {\left(D_{\Phi_{1}}\circ P_{\alpha_{\min}}\right)}^{c}\right)}^{c}\right)}^{*}\circ D_{\Phi_{2}}{]}^{c}\\ \; & {\circ E)}^{c})\left(s\right) \end{array}$$

The proof is placed in Appendix A.

$||E\left(=k\right)|{|}_{max}\left(s\right)$ is the skep *k* instantaneous expected cost of the path that starts from state *s* under the maximum scheduler.
(9)$$\begin{array}{cc} \; & ||E\left(=0\right)|{|}_{max}\left(s\right)=0,\\ \; & ||E\left(=k\right)|{|}_{max}\left(s\right)=\left(P_{\alpha_{\max}|\max|}^{k}\circ E\right)\left(s\right)\cdot C_{T_{\alpha_{\max}}\left(t_{k-1},t_{k}\right)}\left(t_{k-1}\right) \end{array}$$
where $t_{m}=\underset{t_{m}\in S}{argmax}\left(P_{\alpha_{\max}|\max|}^{m}\left(s,t_{m}\right)\right).$

The proof is placed in Appendix A.

$||E\left(=k\right)|{|}_{min}\left(s\right)$ is the skep *k* instantaneous expected cost of the path that starts from state *s* under the minimum scheduler.
(10)$$\begin{array}{cc} \; & ||E\left(=0\right)|{|}_{min}\left(s\right)=0,\\ \; & ||E\left(=k\right)|{|}_{min}\left(s\right)=\left(P_{\alpha_{\min}|\max|}^{k}\circ E\right)\left(s\right)\cdot C_{T_{\alpha_{\min}}\left(t_{k-1},t_{k}\right)}\left(t_{k-1}\right) \end{array}$$
where $t_{m}=\underset{t_{m}\in S}{argmax}\left(P_{\alpha_{\min}|\max|}^{m}\left(s,t_{m}\right)\right).$

The proof is placed in Appendix A.

$||E\left(\leq k\right)|{|}_{max}\left(s\right)$ is the first *k* steps cumulative expected cost of the path that starts from state *s* under the maximum scheduler.  
(11)$$\begin{array}{cc} \; & ||E\left(\leq 0\right)|{|}_{max}\left(s\right)=Ex_{max}^{s}\left(cost\left[=0\right]\right)=0\\ \; & ||E\left(\leq k\right)|{|}_{max}\left(s\right)\\ \; & =Ex_{max}^{s}\left(cost\left[\leq k\right]\right)\\ \; & =\sum_{i=1}^{k}\left(\left(P_{\alpha_{max}|max|}^{i}\circ E\right)\left(s\right)\cdot C_{\alpha_{i-1}}\left(t_{i-1}\right)\right)\\ \; & =\sum_{i=1}^{k}(\left(P_{\alpha_{max}|max|}^{i}\circ E\right)\left(s\right)\cdot C_{T_{\alpha_{\max}}\left(t_{i-1},t_{i}\right)}\left(t_{i-1}\right)\\ \; & ||E\left(\leq k\right)|{|}_{max}\left(s\right)=\sum_{i=1}^{k}(\left(P_{\alpha_{max}|max|}^{i}\circ E\right)\left(s\right)\cdot C_{T_{\alpha_{\max}}\left(t_{i-1},t_{i}\right)}\left(t_{i-1}\right) \end{array}$$
where $t_{m}=\underset{t_{m}\in S}{argmax}\left(P_{\alpha_{\max}|\max|}^{m}\left(s,t_{m}\right)\right).$

$||E\left(\leq k\right)|{|}_{min}\left(s\right)$ is the first *k* steps cumulative expected cost of the path that starts from state *s* under the minimum scheduler.
(12)$$\begin{array}{cc} \; & ||E\left(\leq 0\right)|{|}_{min}\left(s\right)=Ex_{min}^{s}\left(cost\left[=0\right]\right)=0\\ \; & ||E\left(\leq k\right)|{|}_{min}\left(s\right)\\ \; & =Ex_{min}^{s}\left(cost\left[\leq k\right]\right)\\ \; & =\sum_{i=1}^{k}\left(\left(P_{\alpha_{min}|max|}^{i}\circ E\right)\left(s\right)\cdot C_{\alpha_{i-1}}\left(t_{i-1}\right)\right)\\ \; & =\sum_{i=1}^{k}(\left(P_{\alpha_{min}|max|}^{i}\circ E\right)\left(s\right)\cdot C_{T_{\alpha_{\min}}\left(t_{i-1},t_{i}\right)}\left(t_{i-1}\right)\\ \; & ||E\left(\leq k\right)|{|}_{min}\left(s\right)=\sum_{i=1}^{k}(\left(P_{\alpha_{min}|max|}^{i}\circ E\right)\left(s\right)\cdot C_{T_{\alpha_{\min}}\left(t_{i-1},t_{i}\right)}\left(t_{i-1}\right) \end{array}$$
where $t_{m}=\underset{t_{m}\in S}{argmax}\left(P_{\alpha_{\min}|\max|}^{m}\left(s,t_{m}\right)\right).$

$||E\left(\Phi \right)|{|}_{max}\left(s\right)$ is the cumulative expected cost of the path that starts from state *s* and can reach a state in *F* under the maximum scheduler.
(13)$$\begin{array}{cc} \; & ||E\left(\Phi \right)|{|}_{max}\left(s\right)\\ \; & =Ex_{max}^{s}\left(cost\left[⋄Sat\left(\Phi \right)\right]\right)\\ \; & =\underset{1\leq k\leq |S|-|Sat(\Phi )|}{sup}Ex_{max}^{s}\left(cost\left[\leq k\left(⋄Sat\left(\Phi \right)\right)\right]\right)\\ \; & =\underset{2\leq k\leq |S|-|Sat(\Phi )|}{sup}\left\{Ex_{max}^{s}\left(cost\left[\leq 1\left(⋄Sat\left(\Phi \right)\right)\right]\right),Ex_{max}^{s}\left(cost\left[\leq k\left(⋄Sat\left(\Phi \right)\right)\right]\right)\right\}\\ \; & =\underset{2\leq k\leq |S|-|Sat(\Phi )|}{sup}\{\left(P_{\alpha_{max}|Sat\left(\Phi \right)|\max|}^{}\circ E\right)\left(s\right)\cdot C_{T_{\alpha_{max}}\left(s,t\right)}\left(s\right),\\ \; & \sum_{i=1}^{k-1}\left(\left(P_{\alpha_{max}|Sat\left(\Phi \right)|\max|}^{{}^{\prime }i}\circ E\right)\left(s\right)\cdot C_{T_{\alpha_{max}}\left(t_{i-1},t_{i}\right)}\left(s\right)\right)+\\ \; & \left(\left(P_{\alpha_{max}|Sat\left(\Phi \right)|\max|}^{{}^{\prime }k-1}\circ P_{\alpha_{max}|Sat\left(\Phi \right)|\max|}^{}\circ E\right)\left(s\right)\cdot C_{T_{\alpha_{max}}\left(t_{k-1},t_{k}\right)}\left(t_{k-1}\right))\right\} \end{array}$$
where $t=\underset{t\in Sat(\Phi )}{argmax}\left(P_{\alpha_{\max}|Sat\left(\Phi \right)|\max|}\left(s,t\right)\right)$,

$t_{m}=\underset{t_{m}\in S/Sat\left(\Phi \right)}{argmax}\left({\left(P_{\alpha_{\max}|Sat\left(\Phi \right)|\max|}^{\prime }\right)}^{m}\left(s,t_{m}\right)\right),$ if 1≤m≤k−1,

$t_{m}=\underset{t_{m}\in Sat\left(\Phi \right)}{argmax}(({\left(P_{\alpha_{\max}|\mathrm{Sat}\left(\Phi \right)|\max|}^{\prime }\right)}^{m}\circ P_{\alpha_{\max}|\mathrm{Sat}\left(\Phi \right)|\max|}^{})\left(s,t_{m}\right))$ if m=k.

$||E\left(\Phi \right)|{|}_{min}\left(s\right)$ is the cumulative expected cost of the path that starts from state *s* and can reach a state in *F* under the minimum scheduler.
(14)$$\begin{array}{cc} \; & ||E\left(\Phi \right)|{|}_{min}\left(s\right)\\ \; & =Ex_{min}^{s}\left(cost\left[⋄Sat\left(\Phi \right)\right]\right)\\ \; & =\underset{1\leq k\leq |S|-|Sat(\Phi )|}{sup}Ex_{min}^{s}\left(cost\left[\leq k\left(⋄Sat\left(\Phi \right)\right)\right]\right)\\ \; & =\underset{2\leq k\leq |S|-|Sat(\Phi )|}{sup}\{Ex_{min}^{s}\left(cost\left[\leq 1\left(⋄Sat\left(\Phi \right)\right)\right]\right),\\ \; & Ex_{min}^{s}\left(cost\left[\leq k\left(⋄Sat\left(\Phi \right)\right)\right]\right)\}\\ \; & =\underset{2\leq k\leq |S|-|Sat(\Phi )|}{sup}\{\left(P_{\alpha_{min}|Sat\left(\Phi \right)|\max|}^{}\circ E\right)\left(s\right)\cdot C_{T_{\alpha_{min}}\left(s,t\right)}\left(s\right),\\ \; & \sum_{i=1}^{k-1}\left(\left(P_{\alpha_{min}|Sat\left(\Phi \right)|\max|}^{{}^{\prime }i}\circ E\right)\left(s\right)\cdot C_{T_{\alpha_{min}}\left(t_{i-1},t_{i}\right)}\left(s\right)\right)+\\ \; & \left(\left(P_{\alpha_{min}|Sat\left(\Phi \right)|\max|}^{{}^{\prime }k-1}\circ P_{\alpha_{min}|Sat\left(\Phi \right)|\max|}^{}\circ E\right)\left(s\right)\cdot C_{T_{\alpha_{min}}\left(t_{k-1},t_{k}\right)}\left(t_{k-1}\right))\right\} \end{array}$$
where $t=\underset{t\in Sat(\Phi )}{argmax}\left(P_{\alpha_{\min}|Sat\left(\Phi \right)|\max|}\left(s,t\right)\right)$,

$t_{m}=\underset{t_{m}\in S/Sat\left(\Phi \right)}{argmax}\left({\left(P_{\alpha_{\min}|Sat\left(\Phi \right)|\max|}^{\prime }\right)}^{m}\left(s,t_{m}\right)\right),$ if 1≤m≤k−1,

$t_{m}=\underset{t_{m}\in Sat\left(\Phi \right)}{argmax}(({\left(P_{\alpha_{\min}|\mathrm{Sat}\left(\Phi \right)|\max|}^{\prime }\right)}^{m}\circ P_{\alpha_{\min}|\mathrm{Sat}\left(\Phi \right)|\max|}^{})\left(s,t_{m}\right))$ if m=k.

According to (1)–(14), we provide three algorithms to solve the problem of FCTL model checking with cost. Algorithm 1 is used to catch some values of some parameters which would be used to calculate the cost operators. Algorithm 2 is used to calculate the truth values of the formal FCTL state formulas. Algorithm 3 is used to calculate the cost operators.
**Algorithm 1** Catch the action**Require**:a state *s*, the first k−1 step transition matrix $P_{\alpha }$, the step *k* transition matrix $P_{\beta }$, action index matrix $T_{\alpha }$.**Ensure**:the state $t_{k-1}$ after k−1 steps transition, the state $t_{k}$ after *k* steps transition, the action of step *k*
$T_{\alpha }\left(t_{k-1},t_{k}\right)$.1:**for**t∈S**do**2:    **if** $P_{\alpha }^{k-1}\left(s,t\right)>0$ **then**3:        $t_{k-1}⇐t$4:    **end if**5:    **if** $\left(P_{\alpha }^{k-1}\circ P_{\beta }\right)\left(s,t\right)>0$ **then**6:        $t_{k}⇐t$7:    **end if**8:**end for**9:**return**$t_{k-1},t_{k},T_{\alpha }\left(t_{k-1},t_{k}\right)$

Algorithm 1 is proposed to get the action α and states $s_{k-1}$ and $s_{k}$ in transition $s_{k-1}\overset{\alpha }{⟶}s_{k}$ which are used in the computing of cost operators. By the definition of $P_{\alpha |max|}$, we can use the $P_{\alpha }^{k-1}\left(s,t\right)>0$ to be the determined condition of which is the successor state.
**Algorithm 2** Calculating the formal FCTL state formula**Require**:a FDPC $M_{f}$, a FCTL state formula Φ.**Ensure**:the truth value of ∣∣Φ∣∣(s).1:**if**Φ=true**then**2:    **return** ${\left(1\right)}_{s\in S}$3:**end if**4:**if**Φ=a∈AP**then**5:    **return** ${\left(||a||\left(s\right)\right)}_{s\in S}$6:**end if**7:**if**Φ=¬Φ**then**8:    **return** ${\left(1-||\Phi ||\left(s\right)\right)}_{s\in S}$9:**end if**10:**if**$\Phi =\Phi_{1}\wedge \Phi_{2}$**then**11:    **return** ${\left(||\Phi_{1}||\left(s\right)\wedge ||\Phi_{2}||\left(s\right)\right)}_{s\in S}$12:**end if**13:**if**Φ=∃◯Φ**then**14:    **return** $P_{Adv}\circ P_{\Phi }$15:**end if**16:**if**Φ=∀◯Φ**then**17:    **return** ${\left({\left(P_{Adv}\circ D_{\Phi }\right)}^{c}\circ E\right)}^{c}$18:**end if**19:**if**$\Phi =\exists \Phi_{1}⊔\Phi_{2}$**then**20:    **return** ${\left(D_{\Phi_{1}}\circ P_{Adv}\right)}^{*}\circ P_{\Phi_{2}}$21:**end if**22:**if**$\Phi =\forall \Phi_{1}⊔\Phi_{2}$**then**23:    **return** $P_{\Phi_{2}}\cup ([{\left({\left(D_{\Phi_{1}}\circ P_{Adv}\right)}^{c}\circ D_{S}\right)}^{c}\circ {\left({\left(D_{S}\circ {\left(D_{\Phi_{1}}\circ P_{Adv}\right)}^{c}\right)}^{c}\right)}^{*}$ $\circ D_{\Phi_{2}}{]}^{c}{\circ E)}^{c}$24:**end if**

Algorithm 2 is proposed to calculate the quantitative possibility of state formula by matrix operations based on (1)–(8).

Algorithm 3 is proposed to calculate the cost operators by matrix operations based on (9)–(14).

Now let us analyze the time complexities of our algorithm. We would see the three algorithms as one algorithm and analyze it.

Under the scheduler Adv, we can recursively calculate the truth value of ∣∣Φ∣∣(s) in step ∣Φ∣, which is the number of the sub-formula of which is recursively defined as below. If Φ∈AP∪{true}, then ∣Φ∣=1, $|\Phi_{1}\wedge \Phi_{2}|=|\Phi_{1}|+|\Phi_{2}|+1$, ∣¬Φ∣=∣Φ∣+1, ∣∃◯Φ∣=∣Φ∣=1, ∣∀◯Φ∣=∣Φ∣+1, $|\exists \Phi_{1}⊔\Phi_{2}|=|\Phi_{1}|+|\Phi_{2}|+1$, $|\forall \Phi_{1}⊔\Phi_{2}|=|\Phi_{1}|+|\Phi_{2}|+1$, ∣E(=k)∣=1, ∣E(≤k)∣=k and ∣E(Φ)∣=k×(∣Φ∣+1).

The time complexity of calculating the formula $\Phi =a|\Phi_{1}\wedge \Phi_{2}|\neg \Phi$ is only contacted with the size of FDPC $M_{f}$ and Φ, and is O(∣S∣). The time of calculating the formula Φ=E(=k)∣E(≤k) is only contacted with the size of FDPC $M_{f}$ and Φ and *k*, and is O(∣S∣×k). The time of calculating the formula Φ=∃φ∣∀φ is mainly contacted with the time of calculating the transitive closure of $P_{Adv}$, e.g., $P_{Adv}^{*}$. We use the method of literature [30], and the time complexities is $O(|S{|}^{2}\times log|S|)$. The time of calculating the formula Φ=E(Φ) is contacted with the time of catch and the time of matrix multiplication, and is $O(|S{|}^{4})$. Above all, we give the time complexities of our algorithm.
**Algorithm 3** Calculating the cost operators of FCTL**Require**:a FDPC $M_{f}$, step *k*, a FCTL state formula Φ.**Ensure**:the value of E.1:**if**Φ=E(=k)**then**2:    Call algorithm 1, put $s,k,P_{Adv|max|},P_{Adv|max|},T_{Adv}$, get $t_{k-1},t_{k},T_{Adv}\left(t_{k-1},t_{k}\right)$3:    **return** $\left(P_{Adv|\max|}^{k}\circ E\right)\left(s\right)\cdot C_{T_{Adv}\left(t_{k-1},t_{k}\right)}\left(t_{k-1}\right)$4:**end if**5:**if**Φ=E(≤k)**then**6:    **for** i⇐1 to *k* **do**7:        Call algorithm 1, put $s,i,P_{Adv|max|},P_{Adv|max|},T_{Adv}$, get $t_{i-1},t_{i},T_{Adv}\left(t_{i-1},t_{i}\right)$8:        $sum⇐sum+\left(P_{Adv|\max|}^{i}\circ E\right)\left(s\right)\cdot C_{T_{Adv}\left(t_{i-1},t_{i}\right)}\left(t_{i-1}\right)$9:    **end for**10:    **return** sum11:**end if**12:**if**Φ=E(Φ)**then**13:    Call algorithm 1,put $s,1,P_{Adv|Sat(\Phi )|\max|}^{},P_{Adv|Sat(\Phi )|\max|}^{},T_{Adv}$, get $s_{0},s_{1},T_{Adv}\left(s_{0},s_{1}\right)$14:    $sum\left(0\right)⇐\left(P_{Adv|\mathrm{Sat}(\Phi )|\max|}^{}\circ E\right)\left(s\right)\cdot C_{T_{Adv}\left(s_{0},s_{1}\right)}\left(s\right)$15:    **for** i⇐2 to ∣S∣−∣Sat(Φ)∣ **do**16:        **for** m⇐1 to i−1 **do**17:           Call algorithm 1,put $s,m,P_{Adv|Sat(\Phi )|\max|}^{\prime },P_{Adv|Sat(\Phi )|\max|}^{\prime },T_{Adv}$, get $s_{m-1},s_{m},T_{Adv}\left(s_{m-1},s_{m}\right)$18:           $sum\left(i\right)⇐sum\left(i\right)+\left({\left(P_{Adv|\mathrm{Sat}(\Phi )|\max|}^{\prime }\right)}^{m}\circ E\right)\left(s\right)\cdot C_{T_{Adv}\left(s_{m-1},s_{m}\right)}\left(s_{m-1}\right)$19:        **end for**20:        Call algorithm 1, put $s,i,P_{Adv|Sat(\Phi )|\max|}^{\prime },P_{Adv|Sat(\Phi )|\max|},T_{Adv}$, get $s_{i-1},s_{i},T_{Adv}\left(s_{i-1},s_{i}\right)$21:        sum(i)⇐ sum$\left(i\right)+\left({\left(P_{Adv|\mathrm{Sat}(\Phi )|\max|}^{\prime }\right)}^{i-1}\circ P_{Adv|\mathrm{Sat}(\Phi )|\max|}\circ E\right)\left(s\right)\cdot C_{T_{Adv}\left(s_{i-1},s_{i}\right)}\left(s_{i-1}\right)$22:        **if** sum(i)≥sum(0) **then**23:           sum(0)⇐sum(i)24:        **end if**25:    **end for**26:    **return** sum(0)27:**end if**

**Theorem** **1**.
*Let $M_{f}=\left(S,Act,P,I,AP,L,C\right)$ be a finite FDPC, Φ be a FCTL formula and k be a natural number. Then, the time complexities of calculating the truth values or expected cost is $O(size\left(M_{f}\right)\times poly\left(S\right)\times |\Phi |\times k)$, where $size(M_{f})$ is the size of FDPC $M_{f}$, poly(S) is a polynomials of ∣S∣, ∣Φ∣ is the number of the sub-formula of ∣Φ∣, and k is the given natural number.*


## 6. Illustrative Examples

A medical expert system is an intelligent computer system that collects, sorts and analyzes a large number of cases by computer, concentrates on the diagnosis results of medical experts, and diagnoses and treats patients. Because of the different judgment standards of each expert for the degree of the patient’s conditions and the effect of the treatment plan, using a fuzzy system can reflect the operation process of the system closer to the real world. Figure 10, Figure 11, Figure 12 and Figure 13 is a simple medical expert system, in which there are three experts. Each expert gives different treatment plans, which are represented by α, β, γ. The model has four states of the patients, respectively, represented by $s_{0}$, $s_{1}$, $s_{2}$, $s_{3}$. The variables in the state indicate the patient’s health states, which can be divided into *B*(bad), *G*(general), *N*(normal) and *E*(enough). Different experts have a different understanding of these four health conditions. Therefore, we give fuzzy values to the four to show the health of patients. When treatment scheme $\alpha_{i}$ is used in state $s_{i}$, cost $C(s_{i},\alpha_{i})$ will be generated, indicating the treatment cost of the scheme. When using a single treatment scheme, the state transition of patients is shown in Figure 10, Figure 11 and Figure 12. When three experts consult, a complex system is synthesized, as shown in Figure 13. The connecting line with the arrow in the figure indicates transition. The transition possibility is given by the number on the connecting line, and the cost is indicated by the underlined number in the figure. For example, $s_{0}\overset{\alpha ,0.8,\underline{160}}{\to }s_{1}$ indicates that the patient is in state $s_{0}$, using treatment scheme α, then the possibility of transition to state $s_{1}$ is 0.8, and the treatment cost is 160.

(1) $||\exists ◯N|{|}_{max}\left(s_{2}\right)=0.8$, $||\exists ◯N|{|}_{min}\left(s_{2}\right)=0.3$. $||\exists ◯N|{|}_{max}\left(s_{2}\right)=0.8$ is the maximum truth value of that there exists one plan where the patient starts from state $s_{2}$ and becomes normal after one treatment. $||\exists ◯N|{|}_{min}\left(s_{2}\right)=0.3$ is the minimum truth value of that there exists one plan where the patient starts from state $s_{2}$ and becomes normal after one treatment.

(2) $||\forall ◯G|{|}_{max}\left(s_{2}\right)=0.4$, $||\forall ◯G|{|}_{min}\left(s_{2}\right)=0.1$. $||\forall ◯G|{|}_{max}\left(s_{2}\right)=0.4$ is the maximum truth value of all of the plans to satisfy that the patient starts from state $s_{2}$ and becomes general after one treatment. $||\forall ◯G|{|}_{inx}\left(s_{2}\right)=0.1$ is the minimum truth value of all of the plans to satisfy that the patient starts from state $s_{2}$ and becomes general after one treatment.

(3) $||\exists G⊔E|{|}_{max}\left(s_{1}\right)=0.5$, $||\exists G⊔E|{|}_{min}\left(s_{1}\right)=0.4$. $||\exists G⊔E|{|}_{max}\left(s_{1}\right)=0.5$ is the maximum truth value that there exists one plan that the patient starts from state $s_{1}$, keeps general in treatments and becomes enough finally. $||\exists G⊔E|{|}_{min}\left(s_{1}\right)=0.4$ is the minimum truth value that there exists one plan that the patient starts from state $s_{1}$, keeps general in treatments and becomes enough finally.

(4) $||\forall G⊔N|{|}_{max}\left(s_{1}\right)=0.4$, $||\forall G⊔N|{|}_{min}\left(s_{1}\right)=0.1$. $||\forall G⊔N|{|}_{max}\left(s_{1}\right)=0.4$ is the maximum truth value that all of the plans to satisfy that the patient starts from state $s_{1}$, keeps general in treatments and becomes enough finally. $||\forall G⊔N|{|}_{min}\left(s_{1}\right)=0.1$ is the minimum truth value that all of the plans to satisfy that the patient starts from state $s_{1}$, keeps general in treatments and becomes enough finally.

(5) $||E\left(=6\right)|{|}_{max}\left(s_{0}\right)=51.1$, $||E\left(=6\right)|{|}_{min}\left(s_{0}\right)=22$. $||E\left(=6\right)|{|}_{max}\left(s_{0}\right)=51.1$ is the maximum skep 6 instantaneous expected cost of that the patient starts from state $s_{0}$. $||E\left(=6\right)|{|}_{min}\left(s_{0}\right)=22$ is the minimum skep 6 instantaneous expected cost of that the patient starts from state $s_{0}$.

(6) $||E\left(\leq 6\right)|{|}_{max}\left(s_{0}\right)=383.5$, $||E\left(\leq 6\right)|{|}_{min}\left(s_{0}\right)=158$. $||E\left(\leq 6\right)|{|}_{max}\left(s_{0}\right)=383.5$ is the maximum first 6 steps cumulative expected cost of that the patient starts from state $s_{0}$. $||E\left(\leq 6\right)|{|}_{min}\left(s_{0}\right)=158$ is the minimum first 6 steps cumulative expected cost of that the patient starts from state $s_{0}$.

(7) $||E\left(E\right)|{|}_{max}\left(s_{0}\right)=254$, $||E\left(E\right)|{|}_{min}\left(s_{0}\right)=70$. $||E\left(E\right)|{|}_{max}\left(s_{0}\right)=254$ is the maximum cumulative expected cost of that the patient starts from state $s_{0}$ and becomes enough finally. $||E\left(E\right)|{|}_{min}\left(s_{0}\right)=70$ is the minimum cumulative expected cost of that the patient starts from state $s_{0}$ and becomes enough finally.

## 7. Conclusions

This paper provides a polynomial model checking algorithm for the verification of some quantitative properties in fuzzy systems in which in any state a nondeterministic choice and cost between fuzzy sets exist. First, we define a fuzzy decision process model with a cost function. This model can describe the cost consumption and other attributes of a fuzzy system. By introducing the definition of the scheduler, we transmit FDPC into a fuzzy Kripke structure. Next, we give the syntax and semantics of fuzzy computation tree logic with a cost operator to describe the properties. Then, using fuzzy matrix and matrix operations, the quantitative calculation of the computation tree logic model checking on the fuzzy decision process model with the cost is introduced, and the corresponding polynomial time algorithm is proposed.

There are several problems that are worth further study. First, it is interesting to consider the linear temporal logic model checking in FDPC. Second, we would like to extend this method used in this paper to multi-objectives model checking. Finally, we will give some case studies on the methods proposed in this paper.

## Figures and Tables

**Figure 1 entropy-24-01183-f001:**
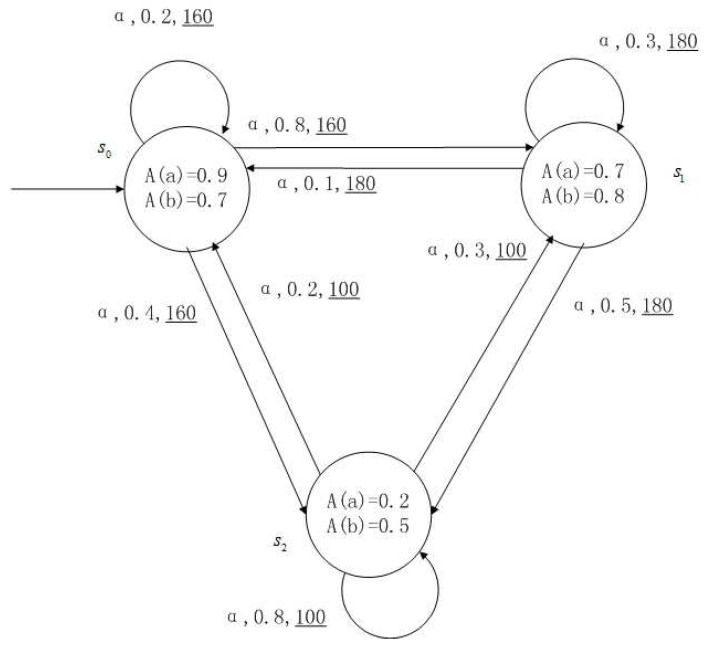
FKS transmitted by α.

**Figure 2 entropy-24-01183-f002:**
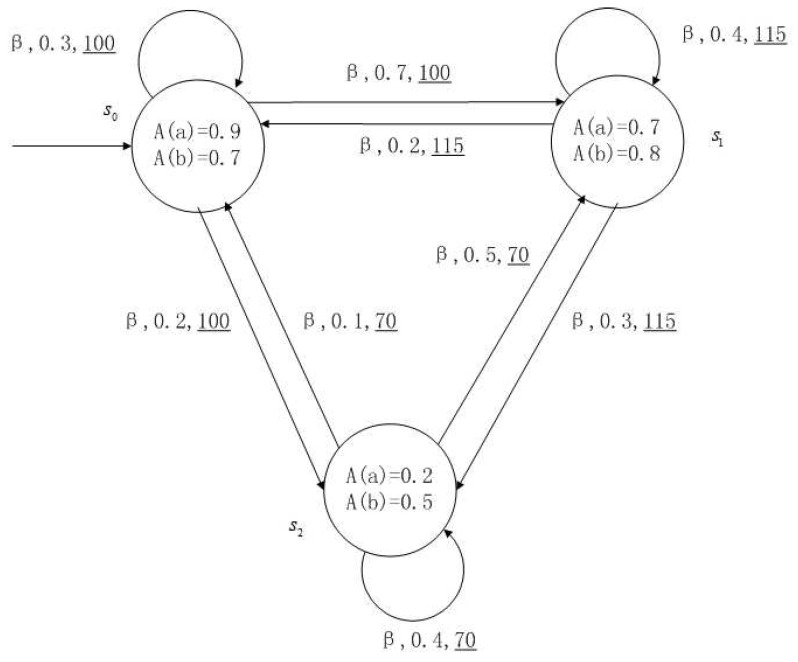
FKS transmitted by β.

**Figure 3 entropy-24-01183-f003:**
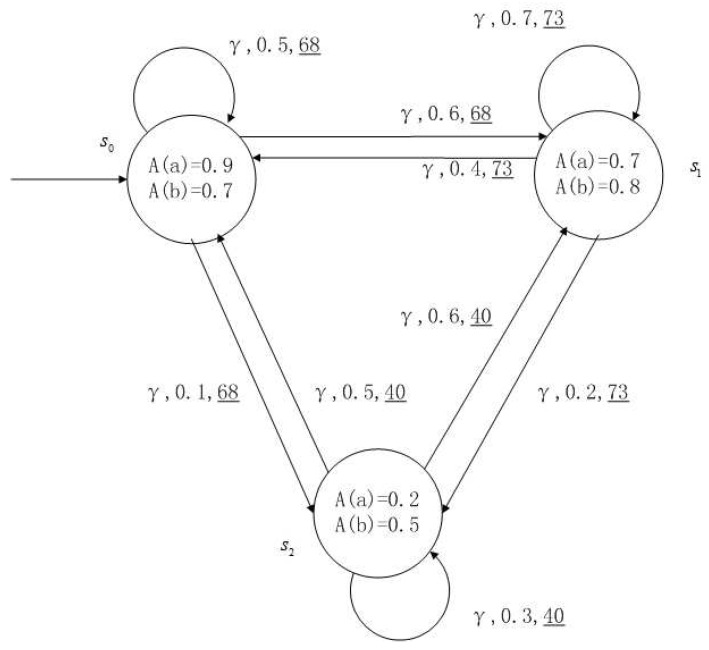
FKS transmitted by γ.

**Figure 4 entropy-24-01183-f004:**
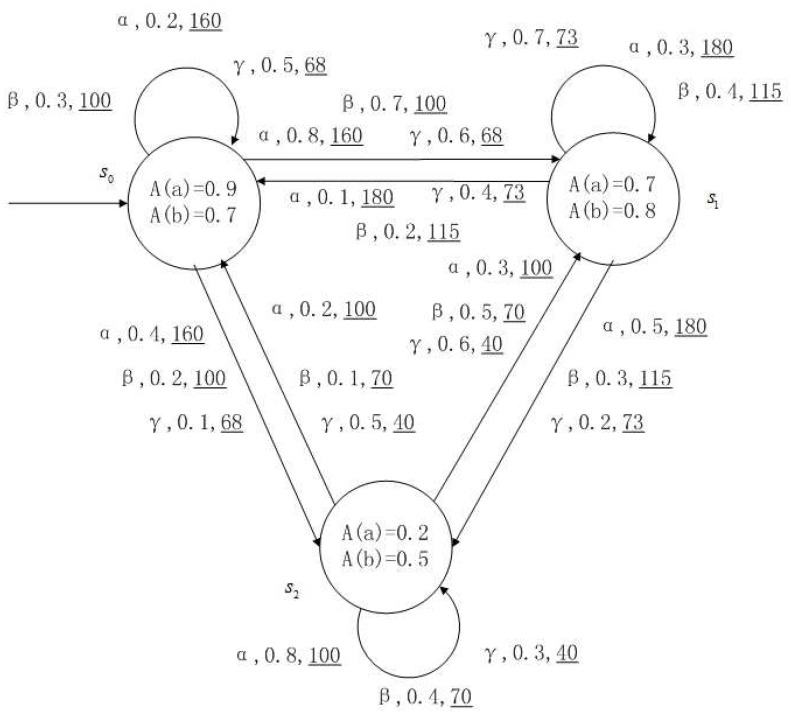
FDPC created by above FKSs.

**Figure 5 entropy-24-01183-f005:**
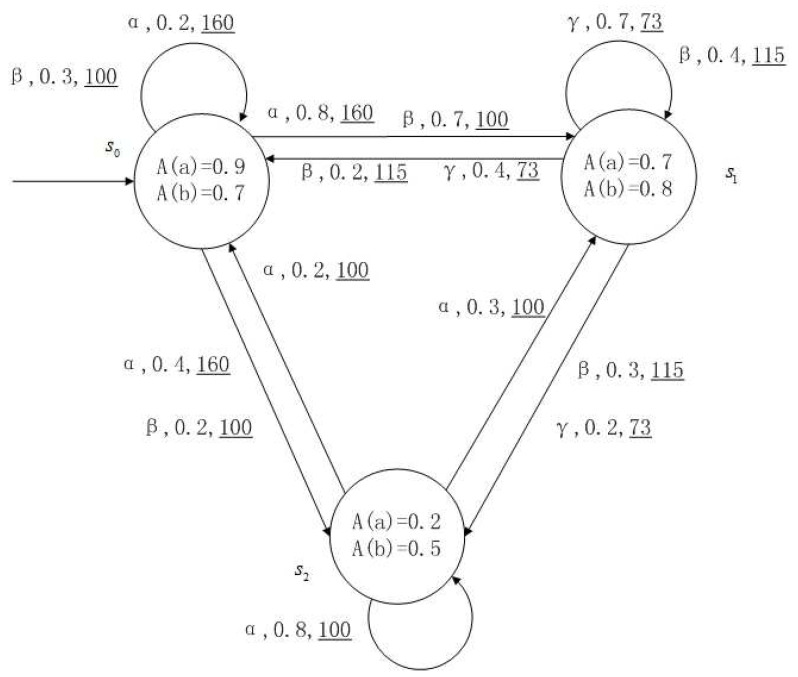
FDPC of Adv.

**Figure 6 entropy-24-01183-f006:**
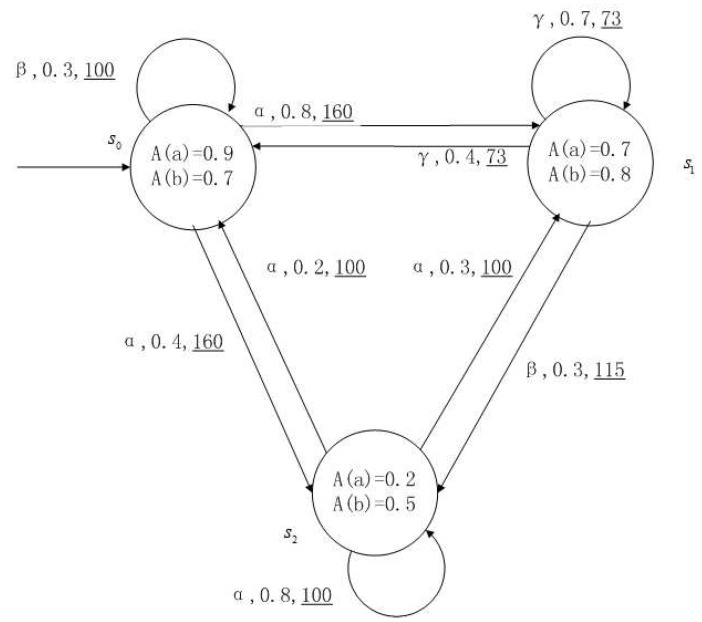
The maximum possibility transition FDPC of Adv.

**Figure 7 entropy-24-01183-f007:**
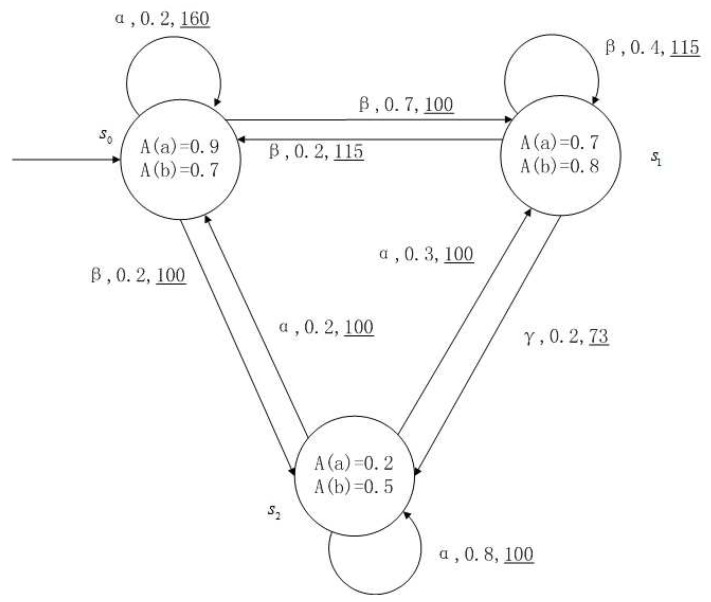
The minimum possibility transition FDPC of Adv.

**Figure 8 entropy-24-01183-f008:**
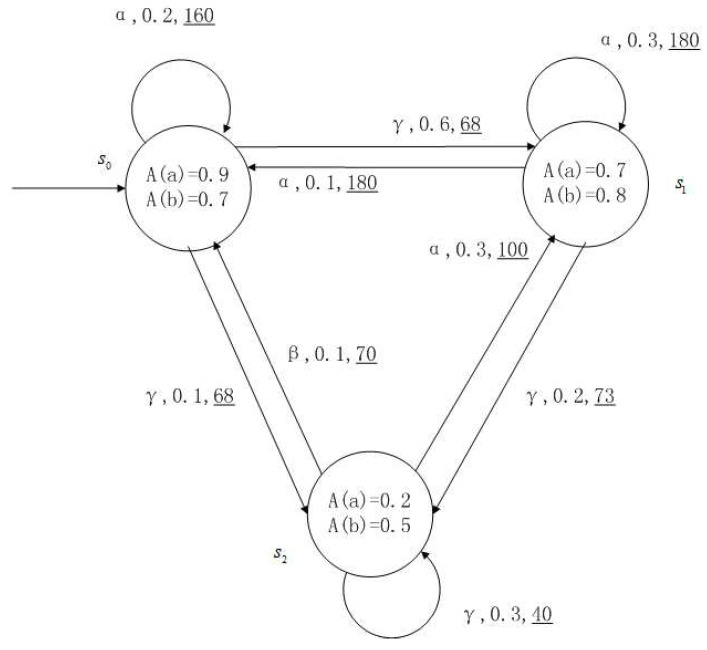
FDPC of $P_{\alpha_{max}}$.

**Figure 9 entropy-24-01183-f009:**
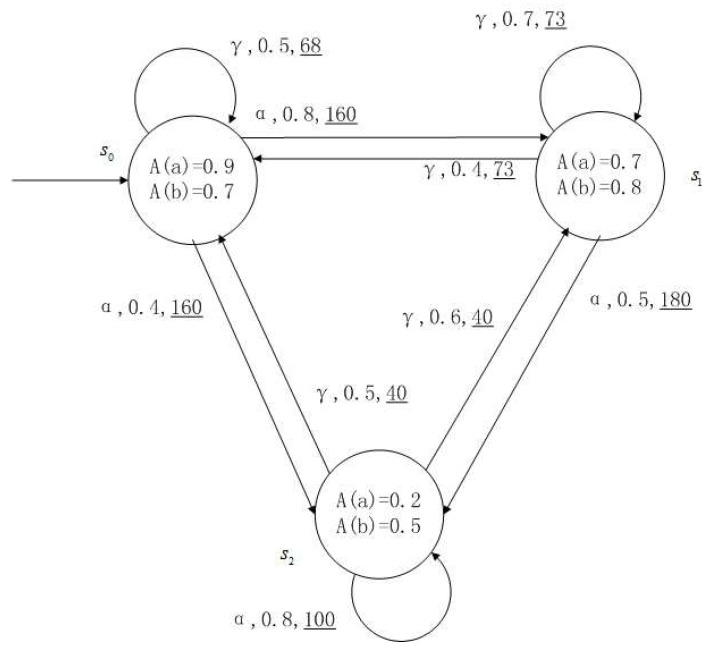
FDPC of $P_{\alpha_{min}}$.

**Figure 10 entropy-24-01183-f010:**
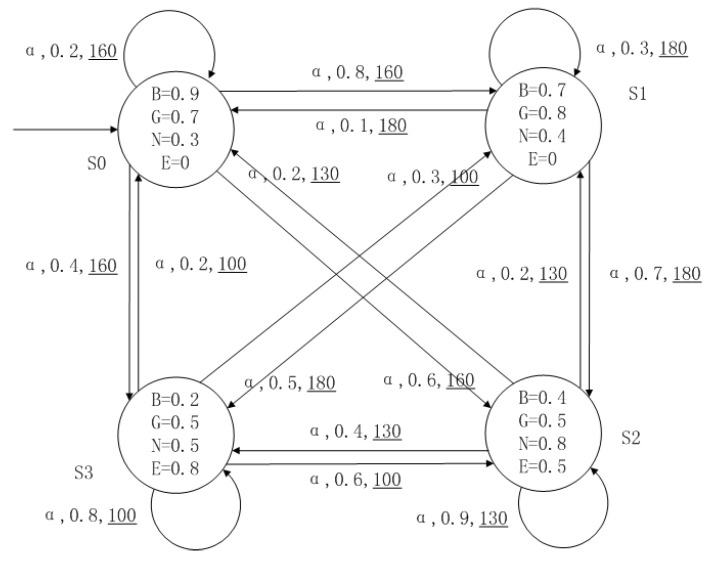
The model of treatment options α.

**Figure 11 entropy-24-01183-f011:**
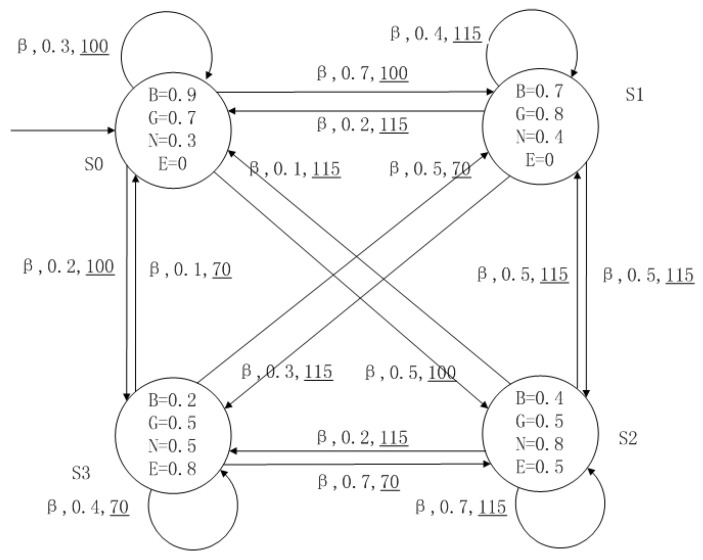
The model of treatment options β.

**Figure 12 entropy-24-01183-f012:**
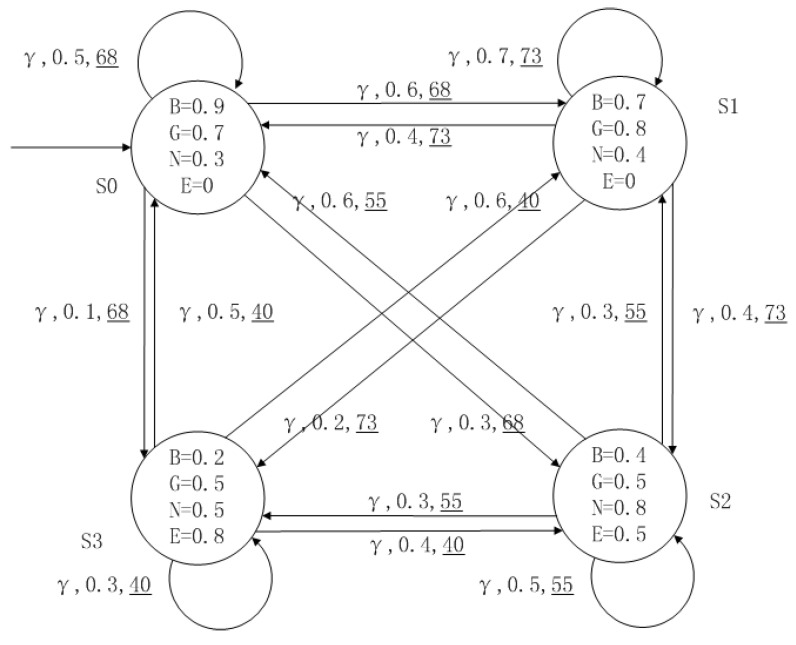
The model of treatment options γ.

**Figure 13 entropy-24-01183-f013:**
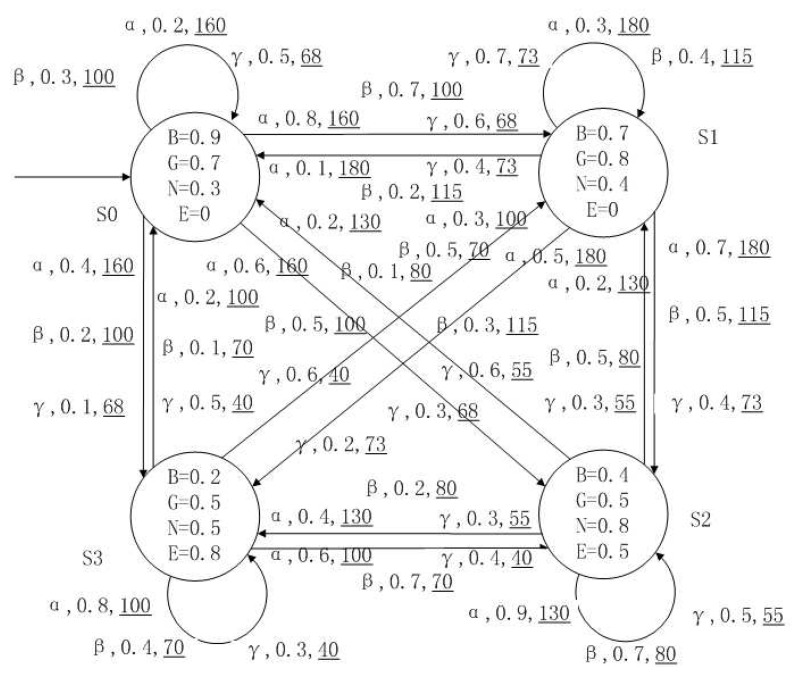
The FDPC of medical expert system.

## Data Availability

Not applicable.

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
