# Peer review of "Model Checking Fuzzy Computation Tree Logic Based on Fuzzy Decision Processes with Cost"

_entropy, 2022, doi:10.3390/e24091183_

Round 1

Reviewer 1 Report

There are some typo corrections which are mentioned on file attached.

Author Response

Dear Reviewer,

We would like to thank you very much for your valuable comments and suggestions that greatly helped to improve our manuscript. We have carefully considered your valuable comments and good suggestions. In the following, we are going to explain how your comments have been taken into full account in the revision. 

Thank you for your good comment. We have made corrections in the updated version of the article.

1) Error: Line 25 uncertainly should be uncertainty

      Correction: Line 27, Original “uncertainly”, now “uncertainty”.

2) Line 40 life , should be life,

      Correction: Line 43, original “life ,”, now “life,”.

3) Line 47 Then , should be Then,

      Correction: Line 57, original “Then ,”, now “Then,”.

4) Line 52 property should be properties

      Correction: Line 62, original “property”, now “properties”.

5) Line 71 “health” should be “healthy”

      Correction: Line 81, original “health”, now “healthy”.

6) Line 78 [0,1] , should be [0,1],

      Correction: Line 89, original “[0,1] ,”, now “[0,1],”.

7) Line 79 [0,1] . should be [0,1].

      Correction: Line 90, original “[0,1] .”, now “[0,1].”.

8) Line 79, 80 X , should be X,

      Correction: Lines 90 and 92, original “X ,”, now “X,”.

9) Line 113 we expend the nation should be we expand the notion

      Correction: Line 125, original “we expend the nation”, now “we expend the notion”.

10) Line 332 which all element is 1 should be with all elements equal to 1.

      Correction: Line 344, original “which all element is 1”, now “with all elements equal to 1”.

Sincerely,

The authors

Reviewer 2 Report

In this article, the authors proposed a fuzzy decision process computation tree logic
model checking method with cost.

Firstly, they introduced a fuzzy decision process
model with cost, which can not only describe the uncertain choice and transition
possibility of systems, but also quantitatively describe the cost of the systems.

Secondly, under the model of fuzzy decision process with cost, they gave the syntax
and semantics of the fuzzy computation tree logic with cost operators.

Thirdly, they studied the problem of computation tree logic model checking for fuzzy decision process with cost, and give its matrix calculation method and algorithm.

This is a novel and good research article. However I have some minor comments for its further improvements.

·      1.  In the introduction section, improve the motivation and the importance of considering the present study over the existing studies.

 2. The conclusion also needs some enhancements. Clearly state your unique research contributions in the conclusion section and point out some potential directions for future work. 

·       3. List of references should also be enhanced by adding some new references related to your research.

Author Response

Dear Reviewer,

We would like to thank you very much for your valuable comments and suggestions that greatly helped to improve our manuscript. We have carefully considered your valuable comments and good suggestions. In the following, we are going to explain how your comments have been taken into full account in the revision.

  1. In the introduction section, improve the motivation and the importance of considering the present study over the existing studies.

â–¸Thank you for your meaningful comment. The motivation and the importance of considering the present study over the existing studies have been added in pages 4 lines 44-51. The details and defects of the existing work are explained, and the motivation of this paper is more strongly put forward by an example.

“For example, consider the disease diagnosis system studied in [14, 19, 26]. Suppose there are multi-steps treatments A and B for a disease, and different treatment needs different costs for each step during the process. The models in literature [14, 19, 26] can only describe the situation that experts have been using treatment A or B during the treatment but cannot describe the situation that experts use A as the first and third steps and use B as the other steps. In addition, the cost of treatments is also unable to represent and verify. ”.

  1. The conclusion also needs some enhancements. Clearly state your unique research contributions in the conclusion section and point out some potential directions for future work.

â–¸Thank you for your great comment. The research contributions and some potential directions for future work in the conclusion section have been added in pages 38 lines 640-642 and 651-655. We use “This paper provides a polynomial model checking algorithm for the verification of some quantitative properties in fuzzy systems in which in any state a nondeterministic choice and cost between fuzzy sets exists” to summarize our contributions. And, the sentences: “There are several problems that are worth further study. First, it is interesting to consider the linear temporal logic model checking in FDPC. Second, we would like to extend this method used in this paper to multi-objectives model checking. Finally, we will give some case studies on the methods proposed in this paper”, are used to point out some potential directions for future work.

  1. List of references should also be enhanced by adding some new references related to your research.

â–¸Thank you for your good comment. We add references 24-26 and cite them, where literature 24 is published in 2019 and is a paper about generalized possibility measures model checking, literature 25 is published in 2022 and is a paper about applications of fuzzy decision, and literature 26 is published in 2021 and is a paper about possibilistic fuzzy linear temporal logic.

Sincerely,

The authors
